# Comparison of Errors Produced by ABA and ITC Methods for the Estimation of Forest Inventory Attributes at Stand and Tree Level in *Pinus radiata* Plantations in Chile

Miguel Ángel Lara-Gómez [1,2], Rafael M. Navarro-Cerrillo [3], Inmaculada Clavero Rumbao [1] and Guillermo Palacios-Rodríguez [2,*]

1    IDAF-Center for Applied Research in Agroforestry Development, Rabanales 21 Science & Technology Park, 14014 Córdoba, Spain
2    Mediterranean Forest Global Change Observatory, Digitalization and Development in Forestry Ecosystems Laboratory, DigiFoR+-ERSAF, Department of Forestry Engineering, University of Cordoba, Campus de Rabanales, Crta. IV km. 396, 14071 Córdoba, Spain
3    Dendrochronology and Climate Change Laboratory, DendrodatLab-ERSAF, Department of Forestry Engineering, University of Cordoba, Campus de Rabanales, Crta. IV km. 396, 14071 Córdoba, Spain
*    Correspondence: gpalacios@uco.es

**Abstract:** Airborne laser scanning (ALS) technology is fully implemented in forest resource assessment processes, providing highly accurate and spatially continuous results throughout the area of interest, thus reducing inventory costs when compared with traditional sampling inventories. Several approaches have been employed to estimate forest parameters using ALS data, such as the Area-Based Approach (ABA) and Individual Tree Crown (ITC). These two methodologies use different information processing and field data collection approaches; thus, it is important to have a selection criterion for the method to be used based on the expected results and admissible errors. The objective of this study was to compare the prediction errors of forest inventory attributes in the functioning of ABA and ITC approaches. A plantation of 500 ha of *Pinus radiata* (400–600 trees ha$^{-1}$) in Chile was selected; a forest inventory was conducted using the ABA and ITC methods and the accuracy of both methods was analyzed. The ITC models performed better than the ABA models at low tree densities for all forest inventory attributes (15% MAPE in tree density—N—and 11% in volume—V). There was no significant difference in precision regarding the volume and basal area (G) estimations at medium densities, although ITC obtained better results for density and dominant height (Ho). At high densities, ABA performed better for all the attributes except for height (6.5% MAPE in N, 8.7% in G, and 8.9% in V). Our results showed that the precision of forest inventories based on ALS data can be adjusted depending on tree density to optimize the selected approach (ABA and ITC), thus reducing the inventory costs. Hence, field efforts can be greatly decreased while achieving better prediction accuracies.

**Keywords:** precision forestry; laser scanning; LiDAR; remote sensing; forest inventory; model-based inference; modeling

## 1. Introduction

Three-dimensional (3D) data obtained from remote sensing in the forestry sector is already a reality that is gradually replacing the traditional tools used in forest inventory [1]. In this respect, remote sensing, which is carried out by freely accessing different spatial and temporal resolution satellite images, along with airborne laser scanning (ALS) technology, is leading to a change in forest inventory [2–4]. ALS technology has been fully validated as a more accurate and less expensive alternative to classic forest inventory. Detailed three-dimensional information regarding the structure of forests is especially useful for forest management inventories (e.g., mean tree height, basal area, and stem volume), logistics

of forest planning, forest fire prevention, structural monitoring, or forest health, among others [5]. ALS technology is being used at different scales, and there have been studies at individual tree, plot, or stand levels allowing forest inventories and monitoring at different spatial resolutions [6]. Moreover, new high-ALS spatial and spectral resolution sensors, algorithms, and computing capacity permit the coverage of large areas, providing accurate information concerning dasometric variables [7].

Two different forest inventory approaches have mainly been applied to date, depending on the ALS point density and the inventory accuracy required: the Area-Based Approach (ABA, [8]) and the Single-Tree Approach (Individual Tree Crown—ITC, [9]). The main characteristic of the ABA methods is that forest inventory attributes are obtained from ALS height metrics at pixel scale, usually by considering a surface area of between 250 and 750 m$^2$. Regression models based on a sample of field plots and ALS observations are then related to obtain mean stand values [10]. The area-based approach is the most common method for predicting forest attributes (e.g., density, dominant height, basal area, and diameter distributions, among others) [8,11]. ABA is faster and makes it technically easier to calculate cloud metrics, in addition to being cost-effective for both computation and laser data collection, thus making ABA the approach most frequently used for inventory-managed forests [11]. However, high-precision georeferenced field data are necessary for adjusted ALS based on regression models to calibrate the link between laser-derived height measurements and forest inventory attributes. Moreover, information cannot be obtained from each individual tree, and it obtaining the distribution of diameter classes is complex. The individual tree crown approach is, on the contrary, based on the segmentation of the ALS-derived canopy height model and produces predictions for each tree inventory attribute, such as the stem diameter and tree height [12,13]. The ITC approach can also be used to estimate individual tree quality attributes, such as the pruning height [14], and, by aggregating individual trees, other dasometric variables, such as the diameter distribution classes, dominant height, or basal area [15,16]. The main limitations of the ITC approach are related to the percentage of trees detected [17], as sometimes only the largest trees are identified [18]. ITC additionally requires the georeferencing of each individual tree on the calibration plots, a high density of ALS points (5–10 points m$^{-2}$), and a longer processing time, thus increasing labor and the cost of forest inventories [17].

The accuracy of the most common forest attributes predicted using the ABA and ITC approaches has been the subject of numerous studies [15,18]. Previous results have shown that both approaches did not significantly differ in average errors when estimating the mean stand characteristics (e.g., average diameter, height, and basal area), whereas ITC produces systematic errors for tree density. For instance, Packalén and Maltamo [19] reported RMSE values of 49.1% with ITC and 27.3% with ABA, while Peuhkurinen et al. [20] and Vastaranta et al. [21] reported only slightly higher RMSE values for basal area and volume estimates with ITC than those with ABA. Additionally, previous studies have pointed out that the ABA and ITC approaches might achieve higher accuracy rates in uniform even-aged forest plantations, thus reducing estimation errors and inventory costs [22]. This is because ITC frequently relies on models of canopy height, and suppressed trees are not identified [23].

However, few of those studies made use of data collected as part of large-area operational inventories and compared the prediction accuracies under different silvicultural conditions. Our objective was, therefore, to compare the performance of the ABA and ITC approaches in predicting the timber volume (V) and basal area (G) of *Pinus radiata* D. Don. plantations in Chile using high-resolution ALS data (15 points·m$^{-2}$). To achieve this objective, (i) a forest inventory was carried out using the ABA and ITC methods and (ii) the ABA and ITC results were compared regarding the volume, tree density, Assman's height, and basal area uncertainty estimation. Our results showed that the precision of forest inventories based on ALS data could contribute to the optimization of the selected approach (ABA and ITC) to reduce inventory costs.

## 2. Materials and Methods

### 2.1. Study Area

The study area comprises 497 ha, including 110 stands of a 19-year-old productive plantation of *Pinus radiata* in the Ñuble Region (Chile, N5939091.572850 m, W793913.723000 m, E799134m, S 5936039m, WGS84 UTM18S) (Figure 1). The plantation area has highly heterogeneous topographic and silvicultural characteristics, with average slopes of between 7 and 14 degrees (maximum of 35 degrees), leading to quite different conditions in terms of microclimates and growth conditions. The elevation ranges between 470 and 830 m above sea level. The pine plantation has different types of vertical vegetation structures, with the inclusion of a shrub layer that is up to 5 m high in some stands, and a high tree canopy layer with living crowns of over 10 m (Table 1). The average tree density is 429 tree ha$^{-1}$ (ranging between 146 and 731 trees ha$^{-1}$, Table 1 and Figure 1). Selective thinning has been conducted to improve the silvicultural characteristics, leading to differences in tree density, with high heterogeneity in the distribution of tree density between stands (Figure 1). Patches of native forests are associated with the stream network, with the presence of *Rubus ulmifolius* Schott and *Aristotelia chilensis* (Molina) Stuntz.

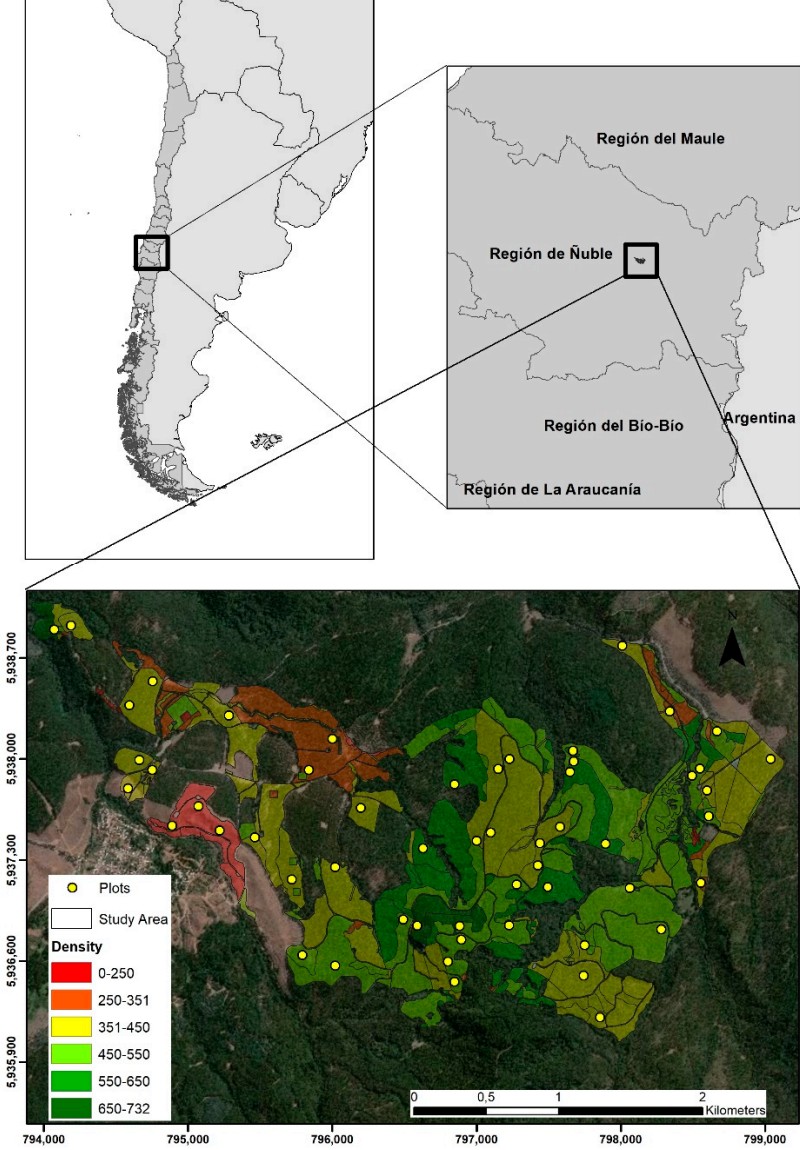

**Figure 1.** Study area and locations of sample plots in the Ñuble Region (Chile, WGS84 UTM18S).

**Table 1.** Silvicultural characteristics of *Pinus radiata* stands in the Ñuble Region (Chile, N = 48 field plots).

| | Mean (min–max) |
|---|---|
| Density (N, trees ha$^{-1}$) | 447 (140–1320) |
| Assman´s dominant height (Ho, m) | 30.57 (15.12–37.25) |
| Basal area (G, m$^2$ ha$^{-1}$) | 26.71 (9.60–46.05) |
| Volume (V, m$^3$ ha$^{-1}$) | 287.52 (122.9–442.2) |

### 2.2. Methodology Framework

Figure 2 describes the workflow followed in this study, highlighting the main steps of data processing used to compare the ABA and ITC approaches to estimate the forest inventory attributes.

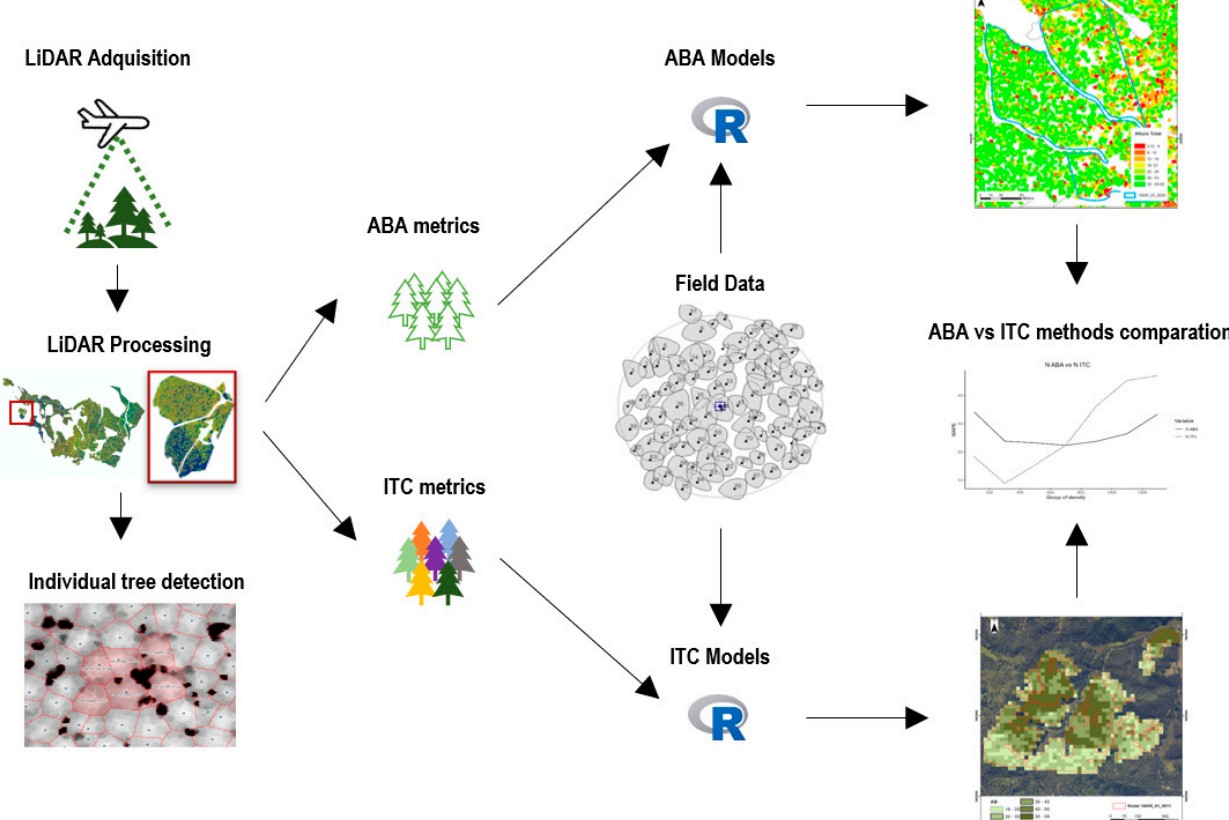

**Figure 2.** Overview of the proposed workflow used to compare the performance of the ABA and ITC approaches for predicting forest inventory attributes.

### 2.3. Field Plot Measurement

Between November and December of 2016, 48 circular plots (r = 12.6 m) were established within the range of the airborne LiDAR strips. The inventory design was based on the forest variability described by the spatial distribution of the 95th percentile of the height of LiDAR cover. The plots were circular with an area of 500 m$^2$ and were distributed by seeking the maximum variability of forest structures (Figure 1). All the trees in each plot with a diameter at breast height (1.3 m above ground level—dbh) greater than or equal to 10 cm were recorded. Field-Map® equipment (Table S1, Supplementary Material, IFER https://fieldmap.cz/, accessed on 14 June 2022) was used. This consisted of an electronic caliper to measure the dbh, and a Ranger ForestPro laser rangefinder and a MapStar compass that allowed the measurement of tree height and sub-metric georeferencing of all trees on the plot, respectively. According to the random errors recorded by post-processing, the planimetric coordinates of the plot centers had an average error of 12 cm.

Field measurements were processed to obtain the forest inventory attributes of each plot. Specifically, the density (N, trees ha$^{-1}$), basal area (G, m$^2$ ha$^{-1}$), Assman´s dominant height (Ho, m), and commercial volume up to 8 cm minimum diameter (V, m$^3$ ha$^{-1}$) were calculated (Table 1). Significant differences between the maximums and minimums of each forest inventory attribute showed that there was high variability among forest stands. High densities were related to the presence of very thin trees on the co-dominant stratum established underneath the canopy of larger trees.

Two trees per plot, one representative of the dominant stratum and one of the co-dominant strata, were additionally selected and cut down. The height, dbh, and diameter for each meter were measured (N = 96, dbh = 17.1–52.2 cm; H = 12.0–35.4 m). This information was then used to generate a local profile equation by parameterizing the coefficients of Riemer´s equation and volume, whose performance was superior for volume estimation in *Pinus radiata* [24].

### 2.4. ALS Data Acquisition and Processing

ALS data acquisition was conducted in October 2016 by Heligraphics Fotogrametría S.L. (Alicante, Spain) using ALS60 laser scanner equipment (Leica-Geosystems AG, Heerbrugg, Switzerland). The resulting point cloud had an average density of 15 pulses m$^{-2}$ (20–12 pulses m$^{-2}$), which were evenly distributed over the entire study surface. The maximum effective FOV was 40° (Table S2, Supplementary Material). LiDAR processing was performed using US Forest Service FUSION/LDV 3.42 software [25] (http://forsys. cfr.washington.edu/fusion/fusionlatest.html, accessed on 23 January 2017). According to the proposed FUSION specifications, the minimum density of 0.5 pulses m$^{-1}$ was the minimum required to produce the 3 m DEM. Proposals referenced in Ruiz et al. [26] were followed to elaborate the Digital Terrain Model (DTM), and a linear prediction-based algorithm was used to create separate filtering processes for the point clouds [27]. Following that, filtered returns were used to generate the DTM (Digital Terrain Model), DSM (Digital Surface Model), and CHM (Canopy Height Model) [28]. The DSM was used to normalize the elevation values of the LiDAR data returns. The FUSION Toolkit was used to extract 43 metrics for each 500 m$^2$ plot. These metrics were used as predictor variables to support the construction of regression models for the estimation of forest inventory attributes by employing the ABA method.

LAStools software was used to generate a Pit-Free CHM with a precision of 0.2 m [29]. Based on the CHM, we segmented the crown outlines with a watershed algorithm [30]. To distinguish the tree top heights from ground and low vegetation, a 2 m threshold was used and an algorithm's height tolerance of 10 cm above 2 m was set. False treetops generated by forked trees or the confusion of dominant branches were subsequently eliminated using the criterion set for minimum distance and height [31] and were then validated by comparing them with the tree positions in the field plots obtained using Field-Map. A polygon was generated around the perimeter of the crown of each tree, and LiDAR metrics were extracted using FUSION/LDV software for each individual tree.

### 2.5. Statistical Modeling

Field forest inventory attributes and ALS metrics were then assessed for multivariate normality and homoscedasticity, after which multivariate linear models were adjusted by employing the ABA and ITC approaches. The ABA models were adjusted to estimate forest inventory attributes (N, G, Ho, and V) using the ALS metrics calculated for each plot as predictor variables. ITC models were fitted to estimate the Ho and dbh using the ALS metrics of each delineated crown as predictor variables. The volume was estimated by applying Riemer´s equation [24] to each tree using the dbh and height estimated using the ITC-ALS models. The models were adjusted by using the stepwise regression method (RCmdr package, [32]) and by taking the Bayesian information criterion (BIC) as the input and output criteria for the variables. To avoid collinearity in the models, variables with a VIF (Variance Inflation Factor) of less than 10 were accepted [33]. The forest attributes at

the plot level using the ITC method were obtained by aggregating the values obtained for each tree, with the aim of making them comparable with the results of the ABA method. In order to compare the results obtained by both methods, the Root-Mean-Square Error (%RMSE) and the Mean-Absolute-Percentage Error (MAPE) were calculated using the residuals between the field data and those estimated by each model.

The suitability of both methods based on forest inventory attributes was studied by analyzing the estimation errors considering the residues in three tree density groups (<400 trees ha$^{-1}$, 406–600 trees ha$^{-1}$, >600 trees ha$^{-1}$), as they are the reference densities in forestry schemes of *P. radiata* plantations in Chile [34].

## 3. Results

### 3.1. Local Riemer´s Equation

Using the tree sample as a basis (N = 96), a local Riemer´s equation was calibrated to estimate the tree volume (Equation (1)). This local Riemer´s equation was used to calculate the volume of each individual tree from its height and dbh measured in the field. These individual volumes were summed per plot to obtain V (m$^3$ ha$^{-1}$).

$$r(h) = \frac{0.429 \cdot dbh}{1 - e^{0.0623 \cdot (1.3 - H)}} + \left(\frac{dbh}{2} - 0.429 \cdot dbh\right) \cdot \left[1 - \frac{1}{1 - e^{0.3975 \cdot (1.3 - H)}}\right] + e^{-0.3975 \cdot h}$$
$$\cdot \left[\frac{\left(\frac{dbh}{2} - 0.429 \cdot dbh\right) \cdot e^{0.5167}}{1 - e^{0.3975 \cdot (1.3 - H)}}\right] - e^{0.0623 \cdot h} \cdot \left[\frac{0.429 \cdot dbh \cdot e^{-0.0623 \cdot H}}{1 - e^{0.0623 \cdot (1.3 - H)}}\right] \tag{1}$$

where *r* = stem radius (cm) at height h (m); *H* = tree height (m); and *dbh* = diameter at breast height (cm).

Figure 3 shows the performance of the parameterized stem profile equation when comparing the observed volume with the predicted volume for each tree (MSE = 0.045 m$^3$).

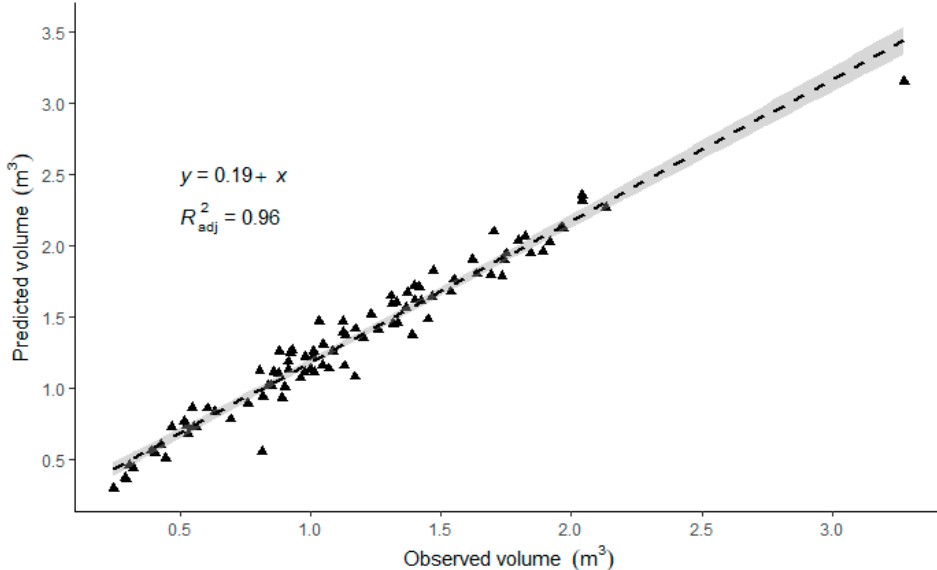

**Figure 3.** Relationship between the volumes (m3) estimated with the parameterized Riemer´s equation and field measurements of Pinus radiata stands in Ñuble Region (South, Chile). The linear model of this relationship (dashed line), the coefficient of determination, and the fitted model equation are shown. The gray shading around the line represents the 95% confidence interval.

### 3.2. ABA and ITC Models for Forest Inventory Attributes

Table 2 shows the best-adjusted stepwise linear models for both the ABA and ITC approaches, which were subsequently used to obtain the estimation errors. The ALS measurements of N, G, Ho, and V were found to be more strongly correlated (R$_{adj}$$^2$ > 0.62) than the ITC measurements. The best regression model for Ho was obtained with the 95th percentile of elevations and the percentage of all returns above the mean as ALS

independent variables ($R_{adj}^2$ = 0.87, RSE = 1.57 m). However, the h and dbh estimation based on the ITC approach obtained a lower coefficient of determination ($R_{adj}^2$ = 0.56, RSE = 4.07 m, and $R_{adj}^2$ = 0.39, RSE = 74.92 mm).

**Table 2.** Adjusted multivariate linear equations for the forest attributes of *Pinus radiata* stands in Ñuble Region (South, Chile) using the Area-Based Approach (ABA) and Individual Tree Crown (ITC) approach. Adjusted models with their predictor variables, adjusted coefficient of determination, residual standard error, F-statistic, and p-value are shown. Density (N), basal area (G), Assman´s dominant height (Ho), and commercial volume of up to 8 cm minimum diameter (V).

| Approach | Forest Attributes | Equation | R$^2$ adj | Residual Standard Error | F-Statistic | *p*-Value |
|---|---|---|---|---|---|---|
| ABA | N (trees ha$^{-1}$) | $-627.278 + 11.356\ V_1 + 2098.761\ V_2$ | 0.62 | 139.70 | 36.37 | <0.001 |
| ABA | G (m$^2$ ha$^{-1}$) | $-4.4228 + 0.2352\ V_3 + 1.0143\ V_4$ | 0.71 | 4.68 | 58.48 | <0.001 |
| ABA | Ho (m) | $-3.36683 + 1.13876\ V_5 + 0.05687\ V_6$ | 0.87 | 1.57 | 160.30 | <0.001 |
| ABA | V (m$^3$ ha$^{-1}$) | $-348.959 + 10.649\ V_1 + 10.796\ V_5$ | 0.79 | 42.62 | 89.81 | <0.001 |
| ITC | h (m) | $3.0805 + 0.9173\ V_5 - 0.279\ V_3 + 0.2169\ V_7$ | 0.56 | 4.07 | 437.00 | <0.001 |
| ITC | dbh (mm) | $177.307 + 6.347\ V_7 - 1.5757\ V_8 + 2.9279\ V_1$ | 0.39 | 74.92 | 83.49 | <0.001 |

$V_1$: Percentage of all returns above the mean; $V_2$: Coefficient of variation of heights; $V_3$: (All returns above 5 m/Total first returns) · 100; $V_4$: Interquartile range of elevations; $V_5$: 95th percentile of elevations of all returns; $V_6$: Percentage of all returns above a mean of 5.00; $V_7$: 60th percentile of elevations of all returns; $V_8$: (All returns above mean/Total first returns) · 100.

### 3.3. Comparison between ABA and ITC Methods

Table 3 shows the errors (%RMSE and MAPE) associated with the analysis of the residues of the ABA and ITC models. In both cases, these statistics were calculated using the residuals obtained by comparing the estimated forest inventory attributes (N, G, Ho, and V) and those obtained in the field plots. The ABA method estimated these attributes by means of the fitted models (Table 2). The ITC method-estimated forest attributes (Table 2) that were aggregated at plot level to obtain forest inventory attributes showed similar values to those obtained by the ABA method.

**Table 3.** Errors (%RMSE and MAPE) obtained for the forest inventory attributes of *Pinus radiata* stands in Ñuble Region (South, Chile) using the Area-Based Approach (ABA) and Individual Tree Crown (ITC) models (see Table 2, n = 48 plots). The lowest error for each variable is highlighted in bold. Density (N, trees ha$^{-1}$), Assman´s dominant height (Ho, m), Basal area (G, m$^2$ ha$^{-1}$), Volume (V, m$^3$ ha$^{-1}$).

| Errors | Forest Attributes | ABA | ITC |
|---|---|---|---|
| % RMSE | N (trees ha$^{-1}$) | 29.89 | **19.68** |
| | G (m$^2$ ha$^{-1}$) | 26.53 | **23.31** |
| | Ho (m) | 5.85 | **4.94** |
| | V (m$^3$ ha$^{-1}$) | 28.38 | **23.03** |
| MAPE | N (trees ha$^{-1}$) | 24.15 | **15.41** |
| | G (m$^2$ ha$^{-1}$) | 20.82 | **19.05** |
| | Ho (m) | 4.51 | **3.56** |
| | V (m$^3$ ha$^{-1}$) | 22.27 | **18.64** |

The estimation models of the forest inventory attributes attained good accuracy with both methods; however, the RMSE and MAPE statistics were better for the ITC method for all variables. To analyze the performance of both methods according to plantation structure, the MAPE distribution of forest inventory variables was plotted in intervals of 200 trees h$^{-1}$ (Figure 4). The ITC models were more precise at low densities (<500 trees ha$^{-1}$ for G and V; <700 trees ha$^{-1}$ for N, and <1000 trees ha$^{-1}$ for Ho), while the ABA models were better at high densities. Table 4 shows the forest inventory attributes for plots belonging to the target interval of tree densities.

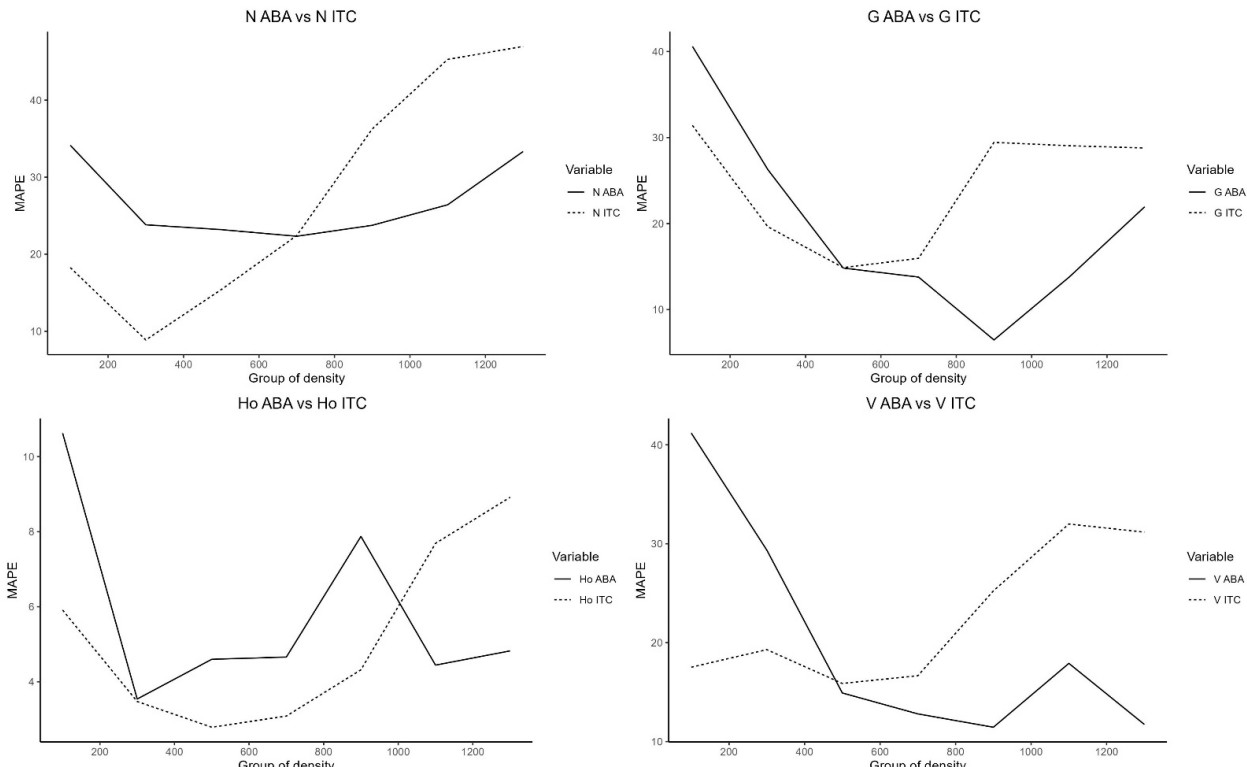

**Figure 4.** Distribution of the mean of the absolute value of the residuals (MAPE,%) in the estimation of the forest inventory attributes of *Pinus radiata* stands in Ñuble Region (South, Chile) using the Area-Based Approach (ABA) and Individual Tree Crown (ITC) models (see Table 2, n = 48 plots) as a function of plot density in intervals of 200 trees ha$^{-1}$. Density (N, trees ha$^{-1}$), Assman´s dominant height (Ho, m), Basal area (G, m$^2$ ha$^{-1}$), Volume (V, m$^3$ ha$^{-1}$).

**Table 4.** Distribution of forest inventory attributes of *Pinus radiata* stands in the Ñuble Region (South, Chile) as a function of plot density in intervals of 200 trees ha$^{-1}$. Density (N, trees ha$^{-1}$), Assman´s dominant height (Ho, m), Basal area (G, m$^2$ ha$^{-1}$), Volume (V, m$^3$ ha$^{-1}$).

| Forest Attributes | N (Trees ha$^{-1}$) | | |
| --- | --- | --- | --- |
| | <400 (n = 24) Mean (Min–Max) | 400–600 (n = 15) Mean (Min–Max) | >600 (n = 9) Mean (Min–Max) |
| N (trees ha$^{-1}$) | 291.7 (140–380) | 480 (420–560) | 806.7 (600–1320) |
| Ho (m) | 31.3 (21.7–36.9) | 30.4 (15.1–37.2) | 29.2 (21.4–33.9) |
| G (m$^2$ ha$^{-1}$) | 20.9 (9.7–30.1) | 29.2 (22.3–40.7) | 33.9 (31.0–46.1) |
| V (m$^3$ ha$^{-1}$) | 251.6 (110.1–403.6) | 319.7 (107.7–442.2) | 328.9 (167.5–399.4) |

Figure 5 shows the RMSE and MAPE values obtained after estimating each forest attribute for each density interval. These errors were calculated by taking the residuals of the plots corresponding to each density interval (Table 4). Similar trends for G and V could be observed, with better results for ITC at low densities, similar results at medium densities, and better results for ABA at high densities. The tree density showed better results for ITC at medium and low densities, and better results for ABA at high densities. In the case of Ho, the results were similar at low densities, improving the ITC results for medium and high densities.

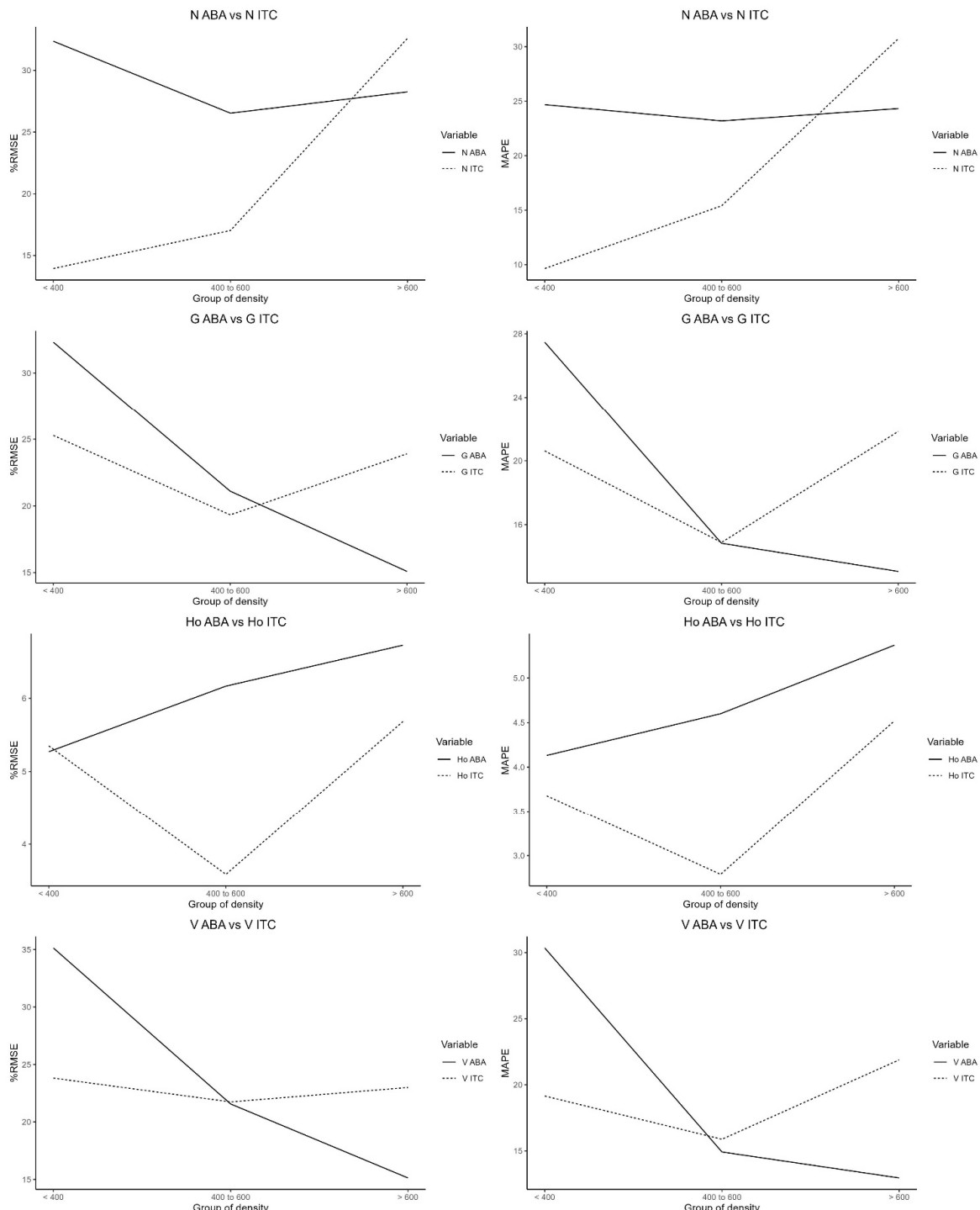

**Figure 5.** Distribution of errors (%RMSE and MAPE) in the estimation of the forest inventory attributes of *Pinus radiata* stands in the Ñuble Region (South, Chile) using the Area-Based Approach (ABA) and Individual Tree Crown (ITC) models (see Table 2, n = 48 plots) as a function of plot density in intervals of 200 trees ha$^{-1}$. Density (N, trees ha$^{-1}$), Assman´s dominant height (Ho, m), Basal area (G, m$^2$ ha$^{-1}$), Volume (V, m$^3$ ha$^{-1}$).

Table 5 presents the statistical performance differences between the ABA and ITC approaches depending on the main forest attributes. In the lowest-density interval (N < 400 trees ha$^{-1}$), the ITC models obtained better results than the ABA models, although this difference was uneven for the various forest inventory attributes. The greatest differences were observed for density and volume (15.0 and 11.2%, respectively, in MAPE),

while the smallest differences were for Ho (0 in %RMSE and 0.5 in MAPE). The ITC models estimated tree density better (improvements of 9.5% RMSE and 7.8 MAPE when compared with the ABA models) at medium densities (N = 400–600 trees ha$^{-1}$), although with less improvement than at a low density. Ho behaved in a similar manner, with a better performance of ITC models when compared with ABA models, although with smaller statistical differences (2.6%RMSE and 1.8 MAPE). G and V performed in a similar manner with both approaches.

**Table 5.** Differences in statistical performance between the ABA and ITC models, quantified as Δ%RMSE (%RMSE ABA–%RMSE ITC) and ΔMAPE (%MAPE ABA–%MAPE ITC), for each forest attribute and each tree density range.

| Forest Attributes | Trees ha$^{-1}$ | Δ%RMSE (ABA–ITC) | Δ MAPE (ABA–ITC) |
|---|---|---|---|
| N (trees ha$^{-1}$) | <400 | 18.5 | 15.0 |
| | 400 to 600 | 9.5 | 7.8 |
| | >600 | −4.3 | −6.4 |
| G (m$^2$ ha$^{-1}$) | <400 | 7.0 | 6.9 |
| | 400 to 600 | 1.8 | −0.1 |
| | >600 | −8.8 | −8.8 |
| Ho (m) | <400 | 0.0 | 0.5 |
| | 400 to 600 | 2.6 | 1.8 |
| | >600 | 1.0 | 0.9 |
| V (m$^3$ ha$^{-1}$) | <400 | 11.3 | 11.2 |
| | 400 to 600 | −0.1 | −1.0 |
| | >600 | −7.9 | −9.0 |

In the high-density range (N > 600 trees ha$^{-1}$), the performance of the models was different with respect to the previous density ranges. Ho obtained the best results for the ITC models, but with a similar performance to the ABA models, with a low difference in the comparison between the two methods (1%RMSE and 0.9 MAPE). The remaining forest attributes obtained better performance in the ABA models when compared with the ITC models, with greater differences than for Ho (6.4 MAPE in N, 8.8 MAPE in G, and 9 MAPE in V).

## 4. Discussion

ALS data have become increasingly implemented for forest inventory [4]. We investigated the ALS base models to predict key forest inventory attributes in *Pinus radiata* plantations in south Chile for optimized forest management. Comparisons of the ABA and ITC approaches for forest inventory are still required to support forest planning at various levels of accuracy and costs, which directly affect the accuracy of the final estimation. Forest inventory attributes derived from diameter and height, and the mean stand characteristics prediction errors were used to compare the methods. The findings of this study showed that both methods could be used to predict important forest characteristics for *P. radiata* plantations in Chile. In our case, the results showed that the best overall prediction accuracy was provided by the ABA method in high-density stands, while the ITC was better at lower tree density. Forest companies that are automating their forest inventory data collection processes through ALS systems should find this strategy to be a practical option. Saving money and speeding up high-accuracy processing are two benefits of reducing field effort, particularly plot data collection [10]. The outcomes acquired from this pilot study have exhibited potential for application in large-scale forest inventories.

### 4.1. Local Riemer´s Taper Function

Numerous stand volume models exist for *Pinus radiata* due to the economic significance of plantations of this species. Taper equations are adaptable and useful tools for estimating

the total and marketable stem volumes in forest inventories [35]. However, few studies have developed local volume equations when conducting ALS forest inventory. As the first step of this research, a nonlinear mixed modeling approach was used to fit and calibrate a stem taper function for *Pinus radiata* plantations in Chile. The greatest improvement in volume and diameter predictions was seen when calibration was performed with an additional diameter that was measured between 40 and 60 percent of the total tree height. As we considered this to be a crucial question when applying to ALS inventories, we also evaluated how stem taper prediction was affected by the expansion of parameters with random effects. The goodness-of-fit statistics indicated that the mixed model with three random effects performed the best among the candidate models. These results are consistent with those of previous studies using a multilevel mixed effects model to describe *P. radiata*'s volume in Northwest Spain [24]. Together, these results suggest that the type of tree variable relationship to be modeled or the data set used influenced the proportion of variability explained by the volume model.

To determine the optimal volume in ALS inventories, the use of local stand volume models based on mixed-effects models performed better when calibrated with additional diameters taken along sample trees, clearly improving the model performance in terms of predictive ability. These results are in line with those suggested in recent studies to improve the accuracy of taper function for radiata pine in New Zealand [36], which suggested measuring an additional diameter at 50% of the total tree height for radiata pine. Thus, developing local regression models to best describe the volume for *P. radiata* to integrate into ALS-based forest inventories could improve stand-level information.

### 4.2. ALS Metrics and Tree Segmentation

Crown segmentation is a fundamental process in the ITC method [37,38], as the extraction of ALS metrics is performed at the tree crown scale. In this work, the tree segmentation accuracy varied as a result of tree density between low-density plantations (MAPE = 9.65 trees ha$^{-1}$, RMSE = 13.9%) and dense plantations (MAPE = 30.76 trees ha$^{-1}$, RMSE = 32.6%). The accuracy of tree segmentation was similar to that found in previous studies [9,38,39], showing that the results of tree individualization obtained herein were similar to those obtained in operating inventories. The presence of tree–crown groups, if it is treated as an individual tree, likely contributed to the underestimation of tree density, which would then lead to an underestimation of other forest attributes (dbh, G, and tree volume). Although the main issue—inaccurate identification of individual trees—was not resolved in this study, which also used a spatial explanatory variable, it did affect the estimation accuracy of the ITC method [40].

The ALS metrics selected by the stepwise method for the different forest inventory attributes were consistent with the plantation structure influenced by horizontal cover (N, G, or dbh) and were explained by coverage metrics, such as the "Percentage of all returns above mean". On the other hand, the height attributes (Ho and h) were explained principally by the height percentile. Volume proved to be highly dependent on both height (95th percentile of elevations of all returns) and horizontal forest structure (percentage of all returns above the mean). The variables selected by the models used in this work coincide with those recommended in previous studies [41]. The improved forest inventory estimation through the use of individual tree and area-based methods emphasized the significance of dominant and co-dominant trees in LiDAR metrics and placed a greater emphasis on the upper canopy LiDAR points [42]. Additional independent variables (coefficient of variation of heights; interquartile range of elevations; 95th percentile, and 60th percentile) provide accurate estimates of the total forest inventory variables, as has occurred in other studies [43,44].

Moreover, the question of whether high-density data are required when selecting the ABA or ITC methods to obtain precise results at the plot scale is a major limitation of using ALS for forest inventory [45,46]. In this study, we used high-density ALS data (15 pulses m$^{-2}$), which have become the standard for monitoring large areas of intensive

pine plantations in Chile. When using statistical height metrics as predictor variables, numerous studies have reported that low plot pulse densities (2–5 pulses m$^{-2}$) have no adverse effects on the quality of forest variables estimation [47,48]. However, high-density ALS data (>10 pulses m$^{-2}$) are still required for the individual tree detection method, which improves regression equations by linking ALS data to field observations, as shown by our findings.

### 4.3. ABA and ITC Models for Forest Inventory Attributes

The multivariate stepwise equations adjusted for ALS metrics showed different performances for dasometric (H, G, Ho, and V) or dendrometric (h and dbh) sets of variables. Table 2 shows that the linear equations to estimate forest attributes based on the ITC method had lower coefficient of determination values than the ABA method. However, when these attributes were summarized at the plot level, the ITC estimations showed better performances than the ABA models (Table 3). This may indicate that there was an error compensation in the estimates of h and dbh in individual trees when they were summed at the plot level.

The ITC models performed better for all the forest attributes when all plots were analyzed together, without distinction of densities (Table 3). However, upon analyzing the performance of the models as a function of tree density, the attribute estimations were affected by this variable in both models, and the ABA models reduced the errors with respect to the ITC models at high tree densities. However, the ITC accuracy was lower at high tree density. Packalén et al. [49] reported similar results in boreal managed forests, with a negative bias of 37% when estimating tree density using the ITC approach, whereas the mean height and volume were estimated with very little bias using both the ITC and ABA models. The research by Vastaranta et al. [50] was also similar in that, despite using a pulse density for ITD that was also relatively low (1.08 pulses m$^{-2}$), they reported only slightly higher RMSEs for the basal area and volume estimates with ITC than with ABA.

The performance of the ITC models is strongly influenced by crown delineation owing to omission errors [9,51]. It is, therefore, to be expected that ITC models will yield higher errors in more structurally complex plantations owing to the higher density with more transitional trees. This implies the presence of more compact and poorly defined crowns, and many trees present in the dominated stratum, which can negatively impact the performance of the individualization process. This effect was more evident when analyzing the models' residuals in the three typical tree-density intervals used in silvicultural schemes in Chile [34]. Many other studies have demonstrated lower detection rates in co-dominant trees because of inadequate point cloud representativeness and overstore obscuration [52,53]. However, as these trees account for a low percentage of the total volume, the error of the volume did not increase with a higher tree density, as occurred with the other forest variables, which indicated the presence of more trees with a similar height. In the low-density range (N < 400 trees ha$^{-1}$), the ITC models performed better for all the forest inventory attributes, and crown delimitation errors were lower, reducing the errors resulting from tree identification. There were non-significant differences between models for V and G estimation at medium density (400 < N < 600 trees ha$^{-1}$), although N and Ho performed better for ITC, but with lower statistical differences with respect to ABA. In the case of high densities (N > 600 trees ha$^{-1}$), ABA performed better for all of the inventory attributes except for height. This may have been due to the influence of the individualization error, with a significant impact on the estimation of N, G, and V, but with less importance in the estimation of height [54].

### 4.4. Implications for Forest Management

Previous studies indicate that the ABA approach is the most appropriate when dominant species are key for forest management, because field data acquisition is less demanding, precision is relatively high, and systematic errors are lower [55]. Other studies [9] have shown the non-comparative statistical advantages of both approaches, but ABA is more

appropriate for forest inventories with many unsampled stands or when management decisions need to be made for non-sampled stands. The aforementioned authors also indicated that ITC significantly improved the accuracy, although it was highly influenced by tree segmentation. This coincides with our results, in which the performance of the ABA and ITC depended on tree density and, therefore, on tree segmentation. In this respect, other authors [56] recommend the use of the ITC approach for monospecific stands of a uniform age when most of the trees are in the upper part of the canopy, which favors the detection of individual trees. The higher cost of data collection for ITC inventories must be considered when choosing between approaches because each tree in the plot needs to be georeferenced [9], which requires more field effort and computer work to perform tree segmentation [47]. However, according to Frank et al. [9], ITC allows for better estimation of forest inventory attributes for management that the ABA methods do not estimate, such as the distribution of diameter classes or pruning height, in addition to improvements regarding spatial resolution. Other authors [57] have stated the usefulness of the ITC approach in forests in which the dendrometric information concerning non-dominant trees is important, and it is especially useful in biodiversity studies or in the silvicultural planning of mixed forests. ABA may, therefore, significantly overestimate forestry variables at a low tree density for intensive pine plantations. However, the ABA method can be used as a hierarchical integration approach with which to upscale forest variable estimation from low to high densities when field plot data are spatially limited, as demonstrated in our research. Positively speaking, the computationally faster and simpler ABA method makes it relatively simple to implement [58]. Furthermore, the ABA also can provide estimations of forestry variables that are useful for supporting single-tree-level management decisions, such as forest diameter distribution classes [59,60]. It is, therefore, possible to suggest that ABA-ITC algorithms can be implemented in a framework in order to improve forest inventories and prevent systematic errors at the stand level [61].

The applicability of the results of this work to other *P. radiata* plantations will depend on the silvicultural complexity of the forests with high densities; in our case, with many trees in the dominated stratum and high errors of omission. This situation derives from the method for the selection of trees to be harvested during silvicultural treatments. On many occasions, these trees are selected according to their characteristics as future high-value candidates, such as the straightness of the stem, adequate thickness, absence of branches at the lower part of the stem, etc. This practice causes a non-homogeneous distribution of trees, which favors the development of a dominant and a dominated stratum with difficulties in detection by remote sensing techniques.

In situations of high densities without dominated strata and with homogeneous and well-formed crown structures, it is expected that the individualization process will provide better precision and, therefore, improve the ITC results. We understand that the future improvement of the tree delineation procedures in forests with complex silvicultural characteristics will imply a greater precision of the ITC results, which may change the recommendations for the use of the models for high densities. In this respect, the integration of ALS data with digital photogrammetry and UAV has provided promising results [62], although they cannot solve the problem of omissions owing to the presence of dominant strata.

## 5. Conclusions

Accurate forest inventory results are of great importance for forestry companies, as they allow the economical evaluation of its assets and more efficient planning for a forestry operation. For planted *Pinus radiata* forests, ALS data can be used to accurately predict forest inventory attributes. The ITC-ABA methods used here can be employed in intensive forest plantations. The choice of the inventory method (ABA or ITC) is important in operational forest planning, as it affects the management costs because the ITC method requires a higher density of points, a higher cost of field data acquisition, and a greater analytical effort. The processing of the results of this work shows that it is important to

select the correct unavoidable method according to the necessary dasometric data and silvicultural characteristics of the forest. The ITC models performed better for all of the dasometric variables at low densities (N < 400 trees ha$^{-1}$). Regarding the mean density, there was no significant difference in the precision of the estimation of V and G with either method, although ITC provided better results for N and Ho. In the case of high densities (N > 600 trees ha$^{-1}$), the performance of ABA was better for all variables, with the exception of height. It is, therefore, possible to suggest that ABA-ITC algorithms can be implemented in a framework to improve forest inventories and prevent systematic errors at the stand level. Due to the cost of ALS data and processing, forest inventories may provide multiple advantages from a single product, including inventory assessment, mapping, change detection, and forest health assessment.

**Supplementary Materials:** The following are available online at https://www.mdpi.com/article/10.3390/rs15061544/s1, Table S1. Field-Map® device specifications. Table S2. Lidar survey specifications. Table S3. Lidar metrics.

**Author Contributions:** Conceptualization, M.Á.L.-G., G.P.-R. and R.M.N.-C.; Methodology, M.Á.L.-G., G.P.-R. and R.M.N.-C.; Formal Analysis, M.Á.L.-G.; Investigation, R.M.N.-C., M.Á.L.-G. and G.P.-R.; Resources, G.P.-R.; Data Cleansing, M.Á.L.-G.; Writing—Original Draft Preparation, R.M.N.-C., M.Á.L.-G. and I.C.R.; Writing—Review and Editing, R.M.N.-C., M.Á.L.-G. and G.P.-R.; Project Administration, G.P.-R. All authors have read and agreed to the published version of the manuscript.

**Funding:** This research was funded by the Competitive Call for Contracts in Companies for Doctors in Training to Obtain an Industrial Doctorate 2019 of the University of Córdoba and the Center for Applied Research in Agroforestry Development (IDAF).

**Institutional Review Board Statement:** Not applicable.

**Informed Consent Statement:** Not applicable.

**Data Availability Statement:** The data presented in this study are available on request from the corresponding author. The data are not publicly available owing to institutional restrictions.

**Acknowledgments:** The authors would like to acknowledge and thank the Mediterranean Forest Global Change Observatory for its support through the "Scientific Infrastructures for Global Change Monitoring and Adaptation in Andalusia (INDALO)–LIFEWATCH-2019-04-AMA-01" project, which was co-financed with FEDER funds corresponding to the Pluriregional Operational Programme of Spain 2014–2020.

**Conflicts of Interest:** The authors declare no conflict of interest.

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
