# Peer review of "Comparison of Errors Produced by ABA and ITC Methods for the Estimation of Forest Inventory Attributes at Stand and Tree Level in Pinus radiata Plantations in Chile"

_remotesensing, doi:10.3390/rs15061544_

Round 1
Reviewer 1 Report
It is my pleasure to review the manuscript entitled "Comparison of errors produced by ABA and ITC methods for 2the estimation of forest inventory attributes at stand and tree level in Pinus radiata plantations in Chile". This study try to compare the errors of ABA and ITC methods for the forest inventory attributes and stand level based on ALS data. The authors have done a lot of works about forest inventory at plot level and ALS data processing. But as a scientific research paper, there are several problems that should be solved in this manuscript.
Several major concerns below:
1. Authors mentioned “stand and tree level” at the title, but only provided the results of stand level based on the plot inventory data, unfortunately, many attributes estimation about tree level are missed in this manuscript.
2. The innovation of this manuscript should be highlighted in the abstract and introduction section.
3. The accuracies or errors of forest inventory attributes estimation based on ALS of the previous studies should be added and referenced in the introduction section.
4. Some details and materials about individual tree detection and tree crown segmentation based on ALS data in the ITC method should be provided in the manuscript.
5. What is the relationship between the Local Riemer’s equation (section 3.1) and the ABA and ITC methods?
6. In the discussion part, the discussion is not profound enough, and the analysis of the results lacks references to support the conclusion.
7. The English writing are need improved for the requirement of the scientific paper.
Some more specific comments:
1. Line 89-90: slopes of between 15 and 30%? The unit of the slope is degree.
2. Line 91, replace ma.s.s with meters above sea level.
3. Line 203, section 3.2, why did authors use AGB estimation in the regression models?
4. Table 3: reorganize the table 3 to avoid duplicate headers (ABA and ITC).
5. Line 236: < 1000 trees ha-1 Ho? Maybe ha-1 N.
6. Table 4: replace “400 a 600” with “400-600”.
7. Table 5: reorganize the table 5 to avoid duplicate headers (ABA and ITC).
8. Table 5 and Figure 5 provide the same information, I suggest that authors just keep one of them.
Author Response
General comments:
We appreciate the comments from reviewer, and we have tried to answer all the questions raised in the following paragraphs. All the specific suggestions have been included which have improved the quality of the manuscript.
- 1. Authors mentioned “stand and tree level” at the title, but only provided the results of stand level based on the plot inventory data, unfortunately, many attributes estimation about tree level are missed in this manuscript.
Table 2 shows the results of the models at stand level (N, G, Ho, V) and tree level (h, dbh). For comparison of the results, the individual tree estimates were grouped for each plot (lines 195-199) to obtain a comparable forest attributes values at stand level. The overall results of the paper use these comparable data.
- 2. The innovation of this manuscript should be highlighted in the abstract and introduction section.
We thank the comment, and the abstract has been improved according to the reviewer's recommendations.
L29-33.- The main novelty of this work is that it shows that the precision of the inventories can be adjusted depending on the density, which makes it possible to optimize the choice of ABA and ITC, and the costs of the inventories. Hence, field efforts can be greatly decreased while yet achieving better prediction accuracies than by fitting local models.
- 3. The accuracies or errors of forest inventory attributes estimation based on ALS of the previous studies should be added and referenced in the introduction section.
We appreciate the comment, and the introduction has been improved according to the reviewer's recommendations.
See L81-88.
- 4. Some details and materials about individual tree detection and tree crown segmentation based on ALS data in the ITC method should be provided in the manuscript.
We appreciate the reviewer’s comment, and a details description of individual tree detection and tree crown segmentation based on ALS data in the ITC method has been included.
L174-176.- Based on the CHM, we segmented the crown outlines with a watershed algorithm [30]. To distinguish tree top heights from ground and low vegetation, a 2 m threshold was used and an algorithm's height tolerance to 10 cm above 2 meters was set.
Khosravipour, A., Skidmore, A.K., Isenburg, M., Wang, T., Hussin, Y.A. Generating pit-free canopy height models from air-borne lidar. Photogramm. Eng. Remote Sens.,2014, 80(9), 863-872.
- 5. What is the relationship between the Local Riemer’s equation (section 3.1) and the ABA and ITC methods?
The local Riemer equation was used to calculate the volume of each individual tree from its height and diameter at breast height measured in the field. These individual volumes were summed per plot to obtain V (m3 ha-1). V was used to fit the models and analyze the results. To clarify this point, the following text has been included:
L207-209.- This local Riemer´s equation was used to calculate the volume of each individual tree from its height and diameter at breast height measured in the field. These individual volumes were summed per plot to obtain V (m3 ha-1).
In addition, it has been included in the discussion (L336-359) the advantages of adjusting local equations to obtain better accuracy in LiDAR inventories.
- 6. In the discussion part, the discussion is not profound enough, and the analysis of the results lacks references to support the conclusion.
Discussion section has been revised according to the reviewer's indication, and its length has been increased from 1354 words to 2110. 20 new references have been included.
- 7. The English writing are need improved for the requirement of the scientific paper.
We appreciate the reviewer´s comment regarding English writing. The manuscript has been completely reviewed by a native English editor, Dra. Sally Newton, that University of Córdoba offers to their researchers. We believe the manuscript has improved the clarity and avoided the problems of understanding of the previous version. Also the recommendations of the two reviewers have greatly improved the clarity of the text, which we deeply appreciate.
Specific comments:
- 8. Line 89-90: slopes of between 15 and 30%? The unit of the slope is degree.
We appreciate the reviewer’s suggestion, and slopes have been expressed in degree units.
- 9. Line 91, replace ma.s.s with meters above sea level.
Done.
- 10. Line 203, section 3.2, why did authors use AGB estimation in the regression models?
We thank the reviewer’s comments warning of this error in the text, we eliminate the mention of AGB.
- 11. Table 3: reorganize the table 3 to avoid duplicate headers (ABA and ITC).
We thank the reviewer’s comments, and the Table 3 has been modified according to his/her recommendation.
|
Errors |
Attribute |
ABA |
ITC |
|
% RMSE |
N (trees ha−1) |
29.89 |
19.68 |
|
G (m2ha−1) |
26.53 |
23.31 |
|
|
Ho (m) |
5.85 |
4.94 |
|
|
V (m3ha−1) |
28.38 |
23.03 |
|
|
MAPE |
N (trees ha−1) |
24.15 |
15.41 |
|
G (m2ha−1) |
20.82 |
19.05 |
|
|
Ho (m) |
4.51 |
3.56 |
|
|
V (m3ha−1) |
22.27 |
18.64 |
- 12. Line 236: < 1000 trees ha-1 Ho? Maybe ha-1
We appreciate the reviewer’s comments, but this sentence shows the performance of the model for each forest attribute according to the density group. For better interpretation we modify it as follows:
L262-263.- The ITC models were more precise at low densities (<500 tress ha-1 for G and V; <700 trees ha-1 for N, <1000 trees ha-1 for Ho), while the ABA…
- 13. Table 4: replace “400 a 600” with “400-600”.
We thank the reviewer’s comments, and the Table 4 has been modified according to his/her recommendation.
- 14. Table 5: reorganize the table 5 to avoid duplicate headers (ABA and ITC).
Table 5 has been modified (see answer of caption 15).
- Table 5 and Figure 5 provide the same information; I suggest that authors just keep one of them.
We appreciate this insightful reviewer’s comment. Table 5 has been modified to show complementary and illustrative information on the difference in the size of the difference in the various errors. The authors think it is important that the reader, who could be a forest resource manager, can clearly identify the trend of errors as a function of the density of his forest stand (current or future) through Figure 5, as well as clearly quantify the loss of accuracy when selecting one method or the other (ABA vs. ITC). This allows the manager to evaluate the opportunity cost of an eventual loss of accuracy, versus the increase in investment depending on the method selected, according to the characteristics of the forest stand.

Reviewer 2 Report
This paper compares the errors produced by Area Based Approach and Individual Tree Crown methods for the estimation of forest inventory attributes at stand and tree level in Pinus radiata plantations in Chile. The Introduction is very short and need to be improved before publishing. The methods are well described but the Results are difficult to understand in its present form. Please, improve the clarity of the results especially the explanation of the figures. The Discussion is a little bit of general. Please provide your study specific discussion.
Please find more specific comments in the attached pdf file.

Author Response
General comments:
We appreciate the comments from reviewer, and we have tried to answer all the questions raised in the following paragraphs. All the specific suggestions have been included which have improved the quality of the manuscript.
- The Introduction is very short and need to be improved before publishing.
We thank the reviewer’s suggestion; we have thoroughly revised the discussion section to include his/her suggestion. Also, the bibliography included in the discussion section has been revised to adapt it to the interpretation of the results obtained on this research.
L58-60.- Area-based approach is the most common method for predicting forest attributes (e.g., density, dominant height, basal area, and diameter distributions among others) [8,11].
L77-88.- The accuracy of the most common forest attributes predicted using ABA and ITC approaches has been the subject of numerous studies [15,18]. They discovered that the two approaches did not significantly differ in their average errors when estimating the mean stand characteristics (e.g., average diameter, height, and basal area), whereas ITC produces significant systematic errors for tree density. For instance, Packalén et al. [19] reported RMSE values of 49.1% with ITC and 27.3% with ABA or Peuhkurinen et al. [20] and Vastaranta et al. [21] who reported only slightly larger RMSEs for the basal area and volume estimates with ITC than with ABA. Previous research has pointed out that ABA and ITC approaches might obtain higher accuracy rates in uniform even-aged forest plantations, thus reducing estimation errors and inventory costs [22]. The reason for this is that because ITC frequently relies on models of canopy height, sup-pressed trees are not found and only the largest trees are identified [23].
However, few of those studies made use of data collected as part of large-area operational inventories and compare the accuracy of predictions under different silvicultural conditions.
- 2. The methods are well described but the Results are difficult to understand in its present form. Please, improve the clarity of the results especially the explanation of the figures.
We thank the reviewer’s suggestion; and the results section has been reviewed to improve clarity. Also, the comments of reviewer#1 has contributed to improve the result section.
- 3. The Discussion is a little bit of general. Please provide your study specific discussion.
We thank the reviewer’s suggestion; we have thoroughly revised the discussion section to include his/her suggestion and its length has been increased from 1354 words to 2110. Also, the bibliography included in the discussion section has been revised to adapt it to the interpretation of the results obtained on this research. 20 new references have been included.
Specific comments:
- L60: Please cite tree diameter distribution using ABA approach. There are several studies.
We agree with the reviewer, that it is important to include some reference on tree diameter distribution using ABA approach. Two references have been included:
L58-60.- Area-based approach is the most common method for predicting forest attributes (e.g., density, dominant height, basal area, and diameter distributions among others) [8,11].
Næsset, E. Predicting Forest stand characteristics with airborne scanning laser using a practical two-stage procedure and field data. Remote Sens. Environ.,2002, 80, 88–99
Frank, B., Mauro, F., Temesgen, H. Model-based estimation of forest inventory attributes using lidar: A comparison of the area-based and semi-individual tree crown approaches. Remote Sens.,2020, 12(16), 2525.
- L74: What do you mean by significant systematic errors? how do you differentiate it from the systematic error?
We thank the reviewer’s suggestion; and “significant” has been removed to improve clarity.
- L92: Is it ma.s.l or m.s.l? right the full form.
Done.
- L95: As mention here and in Fig. 1, the mean min and max density are 429, 140 and 731 trees/ha but in Tab 1, the mean, min and max are 447, 140 and 1320 trees/ha. Why is this difference, please explain.
We appreciate the reviewer's comment, warning of this error in the text. Table 1 shows silvicultural characteristics of Pinus radiata measured in the 48 field plots, while 429 (146–731) is the average stand density.
Table 1. Silvicultural characteristics of Pinus radiata stands in the 48 fields plots.
- L107: Please rephrase this sentence.
We appreciate the reviewer's comment, and the sentence has been rewritten according his/her recommendation:
L119-121.- Figure 2 describes the workflow followed in this work, highlighting the main steps of the data processing flow used to compare the ABA and ITC approaches to estimate forest inventory attributes.
- Line 146: Did you eliminate the ground points/echos when calculating the ALS metrics? how much threshold did you use?
For the calculation of the LiDAR metrics the ground points were not removed, however in those metrics where possible a cutoff of 6 metres was applied to avoid the influence of the shrub strata.
- Line 148: How did you generate the DTM, DSM? Please elaborate.
We appreciate the reviewer's comment, and we have improved the explanation of LiDAR data processing.
L163-172.- According to the proposed FUSION specifications, the minimum density of 0.5 pulses m-1 is the minimum required to produce the 3-m DEM. Proposals referenced in Ruiz et al. [26] were followed to elaborate the Digital Terrain Model (DTM), a linear prediction-based algorithm was used to create separate filtering processes for the point clouds [27]. Following that, the filtered returns were used to generate DTM (Digital Terrain Model), DSM (Digital Surface model) and CHM (Canopy Height Mode) [28]. DSM was used to normalize the elevation values of the LiDAR data returns. FUSION Toolkit was used to extract 43 metrics for each 500 m2-plot (Table S3, Supplementary Material). These metrics were used as predictor variables to support the construction of regression models for the estimation of forest inventory attributes by employing the ABA method.
- Line 183: This can be in the method section.
We appreciate the reviewer's suggestion, but we consider the equation to be an important research result, as it was developed from local inventory data.
- Line 192: Figure 3 need more explanation.
We appreciate the reviewer's comment, and the caption of Figure 3 has been improved:
Figure 3. Relationship between the volumes (m3) estimated with the parameterized Riemer´s equation and field measure of Pinus radiata stands in Ñuble Region (South, Chile). The linear model of this relationship (dashed line), the coefficient of determination and the fitted model equation are shown. The gray shading around the line represents the 95% confidence interval.
- Line 232: If the R2 for ABA is high than the ITC, then why the RMSE is better for the ITC than the ABA. Could you please explain/relate these?
Lower RMSE and MAPE values of ITC in Table 3 may indicate that there is an error compensation in the h and dbh estimates on individual trees, when grouped at the plot level.
We appreciate this reviewer comment and incorporate this idea in the discussion as follow:
L364-369.- The presence of tree-crown groups, if it is treated as an individual tree, likely contributed to the underestimation of tree density, which will then lead to an underestimation of other forest attributes (dbh, G and tree volume). Although the main issue—inaccurate identification of individual trees—was not resolved on this study, which al-so was used a spatial explanatory variable, it did affect estimation accuracy of ITC method [39].
- Line 273: Could you explain, why the RMSE (of volume) is low in case of ITC when the stand density is 400-600 but the RMSE is high when the stand density is <400. You have mentioned that if the stand density is low, the ITC performs better but here it seems opposite.
We appreciate this reviewer comment. RMSE is higher at tree density <400 for ITC than at 400-600. But it's still better than ABA, which is what we compared.
- Line 307: How did you find these results? It'll be interesting to see the a table of LiDAR metrics selected by the stepwise method and some ranking based on its importance.
We appreciate this reviewer comment, and the variable selection procedure is included in the following paragraph:
L191-195.- The models were adjusted by using the stepwise regression method (RCmdr package, [32]), and by taking the Bayesian information criterion (BIC) as the input and output criteria for the variables. To avoid collinearity in the models, those equations in which all the variables obtained a VIF (Variance Inflation Factor) of less than 10 were accept-ed [33].
- Line 360: Please correct, Frank et al.
Done.

Round 2
Reviewer 1 Report
The submission has been greatly improved, and I recommend that your manuscript can be accepted. Congratulations.
Author Response
Reviewer#1
General comments:
The submission has been greatly improved, and I recommend that your manuscript can be accepted. Congratulations.
We thank the reviewer’s suggestions; Thanks to his/her contributions, the manuscript has substantially improved.

Reviewer 2 Report
I appreciate the reviewers' efforts in the revised version of the manuscript. However, I still could not understand the term "Figure systematic error" in Line 81. Please correct or write some explanation about it.
Author Response
Reviewer#2
General comments:
- I appreciate the authors' efforts in the revised version of the manuscript.
We thank the reviewer’s suggestions; Thanks to his/her contributions, the manuscript has substantially improved.
- However, I still could not understand the term "Figure systematic error" in Line 81. Please correct or write some explanation about it.
We thank the reviewer’s suggestion, there was a mistake on the expression and has been corrected.
